# Effects of Salt Stress on Growth, Photosynthesis, and Mineral Nutrients of 18 Pomegranate (*Punica granatum*) Cultivars

**Cuiyu Liu [1,2], Xueqing Zhao [1,2] , Junxin Yan [3], Zhaohe Yuan [1,2,*] and Mengmeng Gu [4,*]**

1   Co-Innovation Center for Sustainable Forestry in Southern China, Nanjing Forestry University, Nanjing 210037, China; liucuiyu88@gmail.com (C.L.); zhaoxq402@163.com (X.Z.)
2   College of Forestry, Nanjing Forestry University, Nanjing 210037, China
3   College of Landscape, Northeast forestry university, Harbin 150040, China; yanjunxin@163.com
4   Department of Horticultural Sciences, Texas A&M AgriLife Extension Service, College Station, TX 77843-2134, USA
*   Correspondence: zhyuan88@hotmail.com (Z.Y.); mgu@tamu.edu (M.G.); Tel.: +86-025-85427056 (Z.Y.); +1-979-845-8545 (M.G.)

**Abstract:** Pomegranate (*Punica granatum* L.) is widely grown in arid and semiarid regions, where the salinization may have developed through irrigation. A greenhouse experiment was conducted to investigate NaCl stress on growth, photosynthesis, and nutrients of 18 pomegranate cultivars. One group was irrigated twice a week with a nutrient solution. The other group was watered twice a week with the same nutrient solution and 200 mM NaCl for five weeks. Dry weight, shoot length, new shoot number, root length and number, leaf area, leaf relative water content, and net photosynthesis of salt-treated plants were negatively impacted by salt stress, and there was a significant difference among cultivars. Few foliar damages were observed. Na content of plants significantly increased in all cultivars, while P, S, K, Ca, Mg, Si, Al, Zn content of plants decreased under salt stress. Fe, Mn, and Cu content increased in most cultivars. Pomegranate accumulated supraoptimal Na mostly in roots and transported more K and Ca to shoots, which was attributed to maintaining a higher ratio of K/Na and Ca/Na in the aerial part of plants. Ten of the 18 cultivars were considered salt-tolerant, which would offer a reference for pomegranate cultivation on saline lands.

**Keywords:** pomegranate; salt stress; morphology; nutrients; correlation

## 1. Introduction

Pomegranate (*Punica granatum* L.) is one of the longest cultivated edible fruit trees, with plenty of nutritional and medicinal benefits [1]. Currently, most pomegranate plants are widely grown in arid and semiarid regions, where the salinization may have developed through irrigation [2,3]. Soil salinization has become a considerable limiting factor in agricultural systems [3]. The deleterious effects of salinity on plant growth are associated with osmotic stress, ion toxicity, nutrient deficiency, and/or the combined effects [4,5]. Generally, pomegranate is a salinity tolerant plant [6,7]. Tavousi [6] and Kaveh [8] found that pomegranate was susceptible to water deficit stress and resistant to salt stress. However, whether pomegranate is a salt-tolerant or salt-sensitive plant is still controversial, and there is a considerable difference among cultivars. Bhantana and Lazarovitch [2] found pomegranate was moderately sensitive to salinity and the electric conductivity (EC) of irrigation water for fruit trees should not exceed 2 dS·m$^{-1}$.

Previous studies have shown that salt stress interferes with pomegranate growth and development [2,9]. El-Khawaga et al. [10] reported that seven-year-old "Manfalouty", "Nab-Elgamal",

and "Wonderful" pomegranate had higher reductions in growth, flowering, and yield after irrigating with saline water at an EC of 6.0 dS·m$^{-1}$ than at an EC of 1.8 dS·m$^{-1}$. Naeini et al. [11] reported that saline water (40, 80, or 120 mM NaCl) had reduced stem length, internode length and number, and leaf surface of "Malas Torsh" and "Alak Torsh" pomegranate when compared to control. Reductions were also observed in leaf numbers, dry weight, flowering, and fruit yield in pomegranate "Manfalouty" and "Nab-Elgamal" after irrigation with saline water [12]. The relative water content, electrical conductivity, stomatal conductance, chlorophyll content, and net photosynthetic rate of pomegranate leaves decreased significantly with increased levels of soil salinity [13,14]. Studies with different cultivars found that when the concentration of Na and Cl in pomegranate tissues increased with increasing salinity, the change in Ca and K content varied among different cultivars [15–17].

Most current studies on pomegranate salt tolerance mainly focused on its shoot growth, photosynthesis, leaf Na, and Cl toxicity, and so on. Previous studies found that some pomegranate cultivars showed higher tolerance to salinity than others [11,18]. It is notable that irrigation with saline water can improve salt-tolerant pomegranate fruit qualities such as acidity, sugar content, antioxidant value, and medicinal properties [19]. Hence, salt-tolerant cultivars identification is of great importance in pomegranate breeding and production. The aims of this study are to comprehensively determine the salt tolerance of 18 pomegranate cultivars based on their morphological and physiological responses to salt stress and to investigate the effects of salt stress on macronutrients and micronutrients in different organs of pomegranate plants.

## 2. Materials and Methods

### 2.1. Plant Materials and Treatments

Hard-wood cuttings (diameter, 1.0~2.0 cm) of 18 pomegranate cultivars (Table 1) were collected from Texas A&M AgriLife Research and Extension Center (Uvalde, TX, USA) on 5 February 2019. Four-month-old rooted cuttings were transplanted in 1.5-gallon pots containing substrates (BM2 Berger, Saint-Modeste, QC, Canada) in the greenhouse at Texas A&M University, College Station, TX with 50%~75% relative humidity and 30.0 ± 5.0/25.0 ± 4.0 °C day/night temperature. Uniform plants were chosen and randomly assigned to 5 blocks, each block including 36 plants of 2 plants per cultivar, one plant was control and another plant was treatment. The control plants (CK) were watered with a nutrient solution (50 ppm; 0.3 dS·m$^{-1}$, Peters®Professional Peat Lite Special 20-10-20, Everris Na Inc., Dublin, OH, USA) without additional salts. The salt-treated plants (ST) were irrigated with a nutrient solution (50 ppm) containing 200 mmol/L (mM) NaCl (20.8 dS·m$^{-1}$). All plants were supplied with 200 or 300 mL (depending on the container media moisture) of nutrient solution with or without NaCl twice a week (every 3~4 days) for 35 days. A saucer was placed under containers.

**Table 1.** Dry weight of roots, stems, and leaves, root/shoot ratio and total day weight of pomegranate plants under NaCl stress.

| Cultivar | Root Dry Weight (g) | | Stem Dry Weight (g) | | Leaf Dry Weight (g) | | Root/Shoot Ratio | | | Total Dry Weight (g) | | |
|---|---|---|---|---|---|---|---|---|---|---|---|---|
| | CK | ST | CK | ST | CK | ST | CK | ST | Δ (%) | CK | ST | Δ (%) |
| Al-sirin-nar | 1.21 | 1.33 | 5.73 | 4.97 | 8.08 | 7.71 | 0.09 | 0.10 | 19.96 | 15.02 | 14.01 | −6.74 |
| Dwarf Moy | 1.04 | 0.84 | 4.61 | 2.99 | 9.77 | 8.39 | 0.07 | 0.07 | 1.99 | 15.42 | 12.22 | −20.74 |
| Garnet Sash | 1.62 | 1.02 | 4.96 | 2.52 | 8.42 | 7.13 | 0.12 | 0.11 | −12.76 | 15.00 | 10.67 | −28.86 |
| Kandahar | 1.98 | 1.52 | 9.61 | 4.90 | 12.09 | 9.12 | 0.09 | 0.11 | 19.20 | 23.69 | 15.54 | −34.39 * |
| Kara Bala Miursal | 1.88 | 1.91 | 6.33 | 4.15 | 9.21 | 8.35 | 0.12 | 0.15 | 26.30 * | 17.43 | 14.42 | −17.28 |
| Kara-Kalinskii | 2.69 | 2.08 | 8.49 | 5.72 | 12.50 | 10.29 | 0.13 | 0.13 | −1.04 | 23.68 | 18.09 | −23.59 |
| Kazake | 1.61 | 1.01 | 6.55 | 4.20 | 10.01 | 7.98 | 0.10 | 0.08 | −14.53 | 18.17 | 13.19 | −27.40 |
| Mollar | 2.42 | 1.61 | 10.14 | 5.90 | 11.80 | 9.16 | 0.11 | 0.11 | −3.41 | 24.37 | 16.66 | −31.64 * |
| Pecos | 1.91 | 1.92 | 7.36 | 4.75 | 9.34 | 7.71 | 0.11 | 0.15 | 34.90 ** | 18.61 | 14.38 | −22.74 |
| Red Angel | 0.92 | 0.52 | 3.97 | 2.26 | 8.27 | 6.73 | 0.07 | 0.06 | −22.80 | 13.16 | 9.51 | −27.79 |
| Salavatski | 1.38 | 1.67 | 6.20 | 5.29 | 8.85 | 7.77 | 0.09 | 0.13 | 38.80 ** | 16.44 | 14.72 | −10.44 |
| Sirenevyi | 0.69 | 0.54 | 3.72 | 2.34 | 6.52 | 6.78 | 0.07 | 0.06 | −11.95 | 10.92 | 9.65 | −11.60 |
| Sogidavna | 0.90 | 0.83 | 3.91 | 3.57 | 7.82 | 7.16 | 0.08 | 0.08 | 0.73 | 12.62 | 11.56 | −8.41 |
| Surh-Anor | 2.37 | 1.84 | 8.07 | 5.18 | 8.99 | 7.81 | 0.14 | 0.14 | 1.99 | 19.43 | 14.83 | −23.65 |
| Sweet | 2.23 | 2.13 | 4.38 | 4.57 | 8.82 | 7.87 | 0.17 | 0.17 | 1.36 | 15.43 | 14.57 | −5.59 |
| Sweet Peppermint | 1.19 | 0.98 | 4.47 | 3.34 | 7.27 | 6.92 | 0.10 | 0.10 | −5.51 | 12.93 | 11.24 | −13.08 |
| Vkusanyi | 1.38 | 0.81 | 5.33 | 3.84 | 8.08 | 6.65 | 0.10 | 0.08 | −24.91 * | 14.80 | 11.30 | −23.65 |
| Wonderful | 1.32 | 1.10 | 6.40 | 3.67 | 9.52 | 7.57 | 0.08 | 0.10 | 18.14 | 17.24 | 12.33 | −28.49 |

The values represented the means of 5 replications; CK: untreated group of plants, ST: salt-treated group of plants; Δ calculated as (ST − CK)/CK × 100; multiple comparison was conducted by the Δ values; * and ** indicate significance at 0.05 and 0.01 among cultivars, respectively.

## 2.2. Growth Parameters

Plant height was measured from the soil surface to the top growing point at the beginning and the end of the experiment. At the end of the experiment, the length of the longest root was measured using a tape ruler, and the number of roots (diameter ≥1 mm) and new shoots (length ≥3 cm) were counted (Figure 1). The surface area of 10 leaves at the middle of branches of each plant was measured using Easy Leaf Area software on a camera phone (Redmi Note 5, Xiaomi Corporation, Beijing, China) with a 20 cm distance from leaves [20]. Total leaf areas were estimated according to: leaf area = (green pixel count) × (calibration area/red pixel count), green pixel count = measurement value, calibration area = 4.04, red pixel count = 4 (Supplementary Figure S1). Finally, all leaf, stem, and root samples were oven-dried at 70 °C for 7 days, and their dry weights were determined. The root/shoot ratio was calculated as dividing root day weight with a total dry weight of stem and leaf.

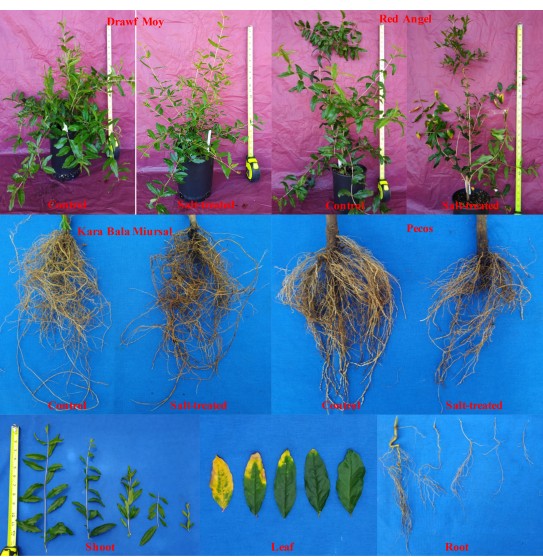

**Figure 1.** The images of shoots and roots of pomegranate plants in control and NaCl-treated groups. The images of multiple shoots, leaves, and roots showed the standards of measurements—new shoots (length ≥ 3 cm), foliar damage (3 leaves on the left with significant yellowing), and the number of roots (diameter ≥ 1 mm).

## 2.3. Leaf Relative Water Content (RWC), SPAD Value and Foliar Salt Damage

At the end of experiment, the relative water content (RWC) of leaves was estimated according to the equation: RWC = (FW − DW)/(TW − DW) × 100%, where FW is fresh weight, TW is turgor weight measured after soaking leaves for 24 h in distilled water at a room temperature, and DW is weight estimated after drying leaves for 48 h at 70 °C. The soil-plant analyses development (SPAD) value of three mature leaves per plant were averaged using a portable chlorophyll meter (SPAD-502, Minolta Camera Co. Ltd., Osaka, Japan). Finally, foliar salt damage was evaluated following the methods described by Sun [14]. Foliar salt damage was observed and the rate was calculated as the percentage of salt damaged plants.

## 2.4. Photosynthetic Parameters

One week before harvest, the leaf net photosynthesis (Pn), stomatal conductance (gs), intercellular $CO_2$ concentration (Ci) and transpiration (E) of each plant were measured using a CIRAS-2 portable photosynthesis system (PP Systems, Amesbury, MA, USA) with an automatic universal PLC6 broadleaf cuvette [14]. Three mature leaves at the middle of each plant branch were chosen for measurement. The parameters in the cuvette were maintained at a leaf temperature of 25 °C, a photosynthetic photon flux of 1000 $\mu mol \cdot m^{-2} \cdot s^{-1}$, and a carbon dioxide concentration of 400 $mmol \cdot mol^{-1}$. Data were recorded

when the environmental conditions and gas exchange parameters in the cuvette became stable. These measurements were taken on sunny days from 9:00 a.m. to 12:00 a.m. and 13:00 p.m. to 16:00 p.m., and plants were well-watered to avoid water stress.

*2.5. Mineral Contents*

Dried root, stem, and leaf samples were finely ground and then filtered using a 40-mesh sieve. The digestion procedure consisted of three steps: (1) a pre-digestion of 0.5 g samples in concentrated $HNO_3$ (5 mL) with the addition of 30% $H_2O_2$ (2 mL) in vessels at room temperature for 30 min; (2) digesting in microwave (MARS 6, CEM, Matthews, North Carolina, USA) on plant setting (~15 min for 195 °C + cooldown); (3) filtering and rinsing into 100 mL volumetric flask with distilled water. Inductively coupled plasma atomic emission spectrometry (ICP-AES; Maxim-III, Applied Research Laboratories, Switzerland) was used to determinate the concentration of phosphorus (P), sulfur (S), potassium (K), calcium (Ca), sodium (Na), magnesium (Mg), iron (Fe), aluminum (Al), silicon (Si), manganese (Mn), copper (Cu), and zinc (Zn) in digest solutions [21].

*2.6. Date Analysis*

The experiment was arraigned in a complete randomized block design with five replications. All the data were analyzed with one-way ANOVA and multiple comparisons with the best treatment (MCB) using Minitab (version 19, LLC, PA, USA). The correlation among variables was analyzed based on the measurement values of the treated and untreated groups and visualized by a "corrplot" package in R [22]. The changes between treatment and control of 12 multivariate parameters were used as salt tolerance indices for hierarchical cluster analysis. The dendrogram of 18 pomegranate cultivars was conducted on these indices using Minitab with the Ward linkage method and squared Euclidian distance [14].

## 3. Results

*3.1. Effects of NaCl Stress on Growth of 18 Pomegranate Cultivars*

As showed in Table 1, most dry weights of roots, stems, and leaves of most cultivars decreased compared to control, and the reductions of plant shoots were much greater than that of roots. The total dry weights of 18 cultivars decreased, between 5.59% and 34.39%, compare to control. "Kandahar" and "Mollar" had the highest reduction of dry weight about 34.39% (8.15 g) and 31.64% (7.71 g), respectively, under NaCl stress (Table 1). An increased root/shoot ratio was observed in 11 cultivars, especially in "Kara Bala Miursal", "Salavatski", "Pecos", and "Vkusanyi" (Table 1, $p < 0.05$ or 0.01).

When compared to control, the shoot height of all cultivars was reduced by NaCl stress (6.45%~59.12%; Figure 1, Table 2), with "Kazake", "Mollar", "Al-sirin-nar", "Kara Bala Miursal", "Kara-Kalinskii", and "Vkusanyi" being significantly inhibited. The reduction of shoot height was below 50% in "Salavatski", "Sweet", "Garnet Sash", and "Kandahar" (Table 2, $p < 0.05$ or 0.01). The numbers of new shoots decreased compared to control (16.78%~64.38%; Table 2), with significant change in 'Red Angel', 'Sweet', 'Sweet Peppermint', 'Garnet Sash', "Wonderful", "Surh-Anor", "Vkusanyi", and "Pecos" (Table 2, $p < 0.05$ or 0.01). The decrease of the average leaf area was about 2.59~6.45 $cm^2$ in 18 pomegranate cultivars compared to control. The leaves of "Dwarf Moy" and "Red Angel" ($p < 0.01$), "Salavatski", "Sweet", "Sweet Peppermint", "Vkusanyi", and "Wonderful" ($p < 0.05$) were significantly inhibited to enlarge and extend (Figure 1, Table 2). The elongation and tillering of pomegranate roots were also repressed by salt stress (Figure 1, Table 2). The root length and number of all cultivars decreased under salt stress (5.22%~43.70% and 3.91%~39.51%, respectively; Table 2). The significant changes in root length of "Red Angel", "Al-sirin-nar", and "Vkusanyi", and root number of "Sogidavna", "Pecos", "Dwarf Moy", "Red Angel", "Garnet Sash" and "Sweet" were observed ($p < 0.01$ or $p < 0.05$).

Table 2. Effects of NaCl stress on shoot height, new shoot number, leaf area, root length, and root number of pomegranate roots, stems and leaves.

| Cultivar | Shoot Height (cm) | | | New Shoot Number | | | Leaf Area (cm$^2$) | | | Root Length (cm) | | | Root Number | | |
|---|---|---|---|---|---|---|---|---|---|---|---|---|---|---|---|
| | CK | ST | Δ (%) | CK | ST | Δ (%) | CK | ST | Δ (%) | CK | ST | Δ (%) | CK | ST | Δ (%) |
| Al-sirin-nar | 31.80 | 16.20 | −49.06 * | 7.00 | 5.20 | −25.71 | 14.52 | 10.88 | −25.03 | 29.00 | 21.60 | −25.52 * | 18.00 | 17.00 | −5.56 |
| Dwarf Moy | 11.00 | 8.30 | −24.55 | 29.80 | 24.80 | −16.78 | 12.67 | 7.23 | −42.91 ** | 23.60 | 19.40 | −17.80 | 23.00 | 16.20 | −29.57 * |
| Garnet Sash | 15.80 | 14.00 | −11.39 | 10.60 | 4.80 | −54.72 * | 15.22 | 11.86 | −22.07 | 30.00 | 24.00 | −20.00 | 20.60 | 15.20 | −26.21 * |
| Kandahar | 16.60 | 14.40 | −13.25 | 29.40 | 16.40 | −44.22 | 11.92 | 9.11 | −23.53 | 27.00 | 21.80 | −19.26 | 24.40 | 18.80 | −22.95 |
| Kara Bala Miursal | 26.80 | 13.60 | −49.25 * | 12.80 | 8.60 | −32.81 | 15.69 | 11.13 | −29.08 | 30.80 | 26.00 | −15.58 | 20.40 | 19.60 | −3.92 |
| Kara-Kalinskii | 27.00 | 13.20 | −51.11 * | 21.40 | 16.60 | −22.43 | 14.06 | 10.25 | −27.07 | 27.20 | 22.40 | −17.65 | 34.20 | 30.20 | −11.70 |
| Kazake | 25.20 | 10.80 | −57.14 ** | 7.60 | 5.40 | −28.95 | 16.88 | 11.91 | −29.42 | 25.80 | 22.60 | −12.40 | 28.20 | 23.80 | −15.60 |
| Mollar | 36.20 | 14.80 | −59.12 ** | 22.60 | 14.60 | −35.40 | 14.40 | 10.39 | −27.87 | 27.00 | 21.20 | −21.48 | 33.60 | 29.40 | −12.50 |
| Pecos | 19.00 | 13.20 | −30.53 | 6.80 | 3.20 | −52.94 * | 17.03 | 12.93 | −24.05 | 33.60 | 28.20 | −16.07 | 24.20 | 16.40 | −32.23 * |
| Red Angel | 14.00 | 9.40 | −32.86 | 14.60 | 5.20 | −64.38 ** | 16.00 | 9.55 | −40.33 ** | 23.80 | 13.40 | −43.70 ** | 12.00 | 8.60 | −28.33 * |
| Salavatski | 18.00 | 16.40 | −8.89 | 6.60 | 4.20 | −36.36 | 14.20 | 9.18 | −35.31 * | 22.40 | 20.80 | −7.14 | 20.40 | 15.40 | −24.51 |
| Sirenevyi | 18.80 | 12.80 | −31.91 | 8.40 | 4.40 | −47.62 | 13.19 | 9.18 | −30.43 | 22.60 | 17.80 | −21.24 | 12.00 | 10.80 | −10.00 |
| Sogidavna | 15.80 | 12.00 | −24.05 | 12.60 | 6.80 | −46.03 | 13.69 | 9.47 | −30.84 | 17.20 | 14.20 | −17.44 | 16.20 | 9.80 | −39.51 ** |
| Surh-Anor | 26.40 | 15.00 | −43.18 | 14.60 | 7.20 | −50.68 * | 11.56 | 8.97 | −22.40 | 29.80 | 24.80 | −16.78 | 24.60 | 19.20 | −21.95 |
| Sweet | 18.60 | 17.40 | −6.45 | 13.60 | 5.60 | −58.82 ** | 16.91 | 11.06 | −34.57 * | 26.80 | 25.40 | −5.22 | 26.20 | 19.40 | −25.95 * |
| Sweet Peppermint | 24.40 | 15.00 | −38.52 | 6.40 | 2.80 | −56.25 * | 15.33 | 9.92 | −35.33 * | 27.40 | 23.20 | −15.33 | 19.00 | 17.20 | −9.47 |
| Vkusanyi | 25.40 | 13.20 | −48.03 * | 7.20 | 3.40 | −52.78 * | 15.57 | 9.96 | −36.08 * | 25.80 | 18.80 | −27.13 * | 17.80 | 13.60 | −23.60 |
| Wonderful | 18.80 | 13.60 | −27.66 | 15.40 | 6.80 | −55.84 * | 14.56 | 9.06 | −37.75 * | 21.00 | 17.20 | −18.10 | 20.40 | 15.20 | −25.49 |

The values represented the means of 5 replications; CK: untreated group of plants, ST: salt-treated group of plants; Δ calculated as (ST − CK)/CK × 100; multiple comparison was conducted by the Δ values; * and ** indicate significance at 0.05 and 0.01 among cultivars, respectively.

## 3.2. Foliar Salt Damage, RWC, and SPAD Value

At the end of the experiment, the foliar salt damage was visually rated and the relative water content (RWC) and soil–plant analyses development (SPAD) values were measured. Salt stress caused a very low level of leaf burn, necrosis, or discoloration on all cultivars (Figure 1), except "Dwarf Moy", "Kandahar", "Kazake", "Mollar", and "Sirenevyi", showing low visual damage. The highest damage (60%) was observed on "Sogidavna" under salt stress ($p < 0.05$). Significant RWC decreases were observed in "Salavatski", "Dwarf Moy", and "Wonderful" (Table 3, $p < 0.05$). We found SPAD values of 11 pomegranate cultivars increased and that of 7 cultivars decreased under salt stress (Table 3, $p < 0.05$).

**Table 3.** Salt damage, relative water content (RWC), and soil–plant analyses development (SPAD) of pomegranate leaves under NaCl stress.

| Cultivar | Salt Damage (%) | | | RWC (%) | | | SPAD | | |
|---|---|---|---|---|---|---|---|---|---|
| | CK | ST | ST−CK | CK | ST | Δ (%) | CK | ST | Δ (%) |
| Al-sirin-nar | 0 | 40 | 40 | 81.07 | 78.17 | −3.58 | 54.38 | 55.92 | 2.84 |
| Dwarf Moy | 0 | 0 | 0 | 83.96 | 71.13 | −15.28 * | 52.68 | 51.93 | −1.43 |
| Garnet Sash | 20 | 60 | 40 | 84.88 | 75.57 | −10.97 | 55.50 | 56.57 | 1.94 |
| Kandahar | 0 | 0 | 0 | 82.85 | 73.29 | −11.54 | 53.92 | 52.55 | −2.53 |
| Kara Bala Miursal | 0 | 20 | 20 | 84.76 | 72.97 | −13.91 | 56.18 | 56.13 | −0.10 |
| Kara-Kalinskii | 0 | 20 | 20 | 84.01 | 72.17 | −14.09 | 57.51 | 58.03 | 0.92 |
| Kazake | 0 | 0 | 0 | 86.05 | 81.64 | −5.12 | 52.41 | 52.77 | 0.70 |
| Mollar | 0 | 0 | 0 | 80.95 | 73.11 | −9.68 | 56.86 | 53.69 | −5.57 * |
| Pecos | 0 | 40 | 40 | 82.31 | 78.56 | −4.56 | 54.33 | 50.80 | −6.50 * |
| Red Angel | 0 | 40 | 40 | 80.10 | 78.80 | −1.62 | 59.17 | 58.57 | −1.01 |
| Salavatski | 0 | 20 | 20 | 87.16 | 71.41 | −18.07 * | 54.61 | 54.64 | 0.06 |
| Sirenevyi | 0 | 0 | 0 | 78.90 | 68.80 | −12.80 | 52.05 | 53.18 | 2.17 |
| Sogidavna | 0 | 60 | 60 * | 77.86 | 67.89 | −12.81 | 54.19 | 54.53 | 0.63 |
| Surh-Anor | 0 | 40 | 40 | 77.57 | 72.98 | −5.92 | 53.68 | 54.23 | 1.02 |
| Sweet | 0 | 40 | 40 | 83.94 | 74.35 | −11.42 | 54.09 | 56.11 | 3.75 |
| Sweet Peppermint | 0 | 40 | 40 | 79.48 | 72.29 | −9.05 | 56.75 | 58.41 | 2.93 |
| Vkusanyi | 20 | 40 | 20 | 81.34 | 75.61 | −7.04 | 55.13 | 55.87 | 1.34 |
| Wonderful | 0 | 20 | 20 | 83.45 | 71.32 | −14.54 * | 57.68 | 56.08 | −2.78 |

The values represented the means of 5 replications; CK: untreated group of plants, ST: salt-treated group of plants; Δ calculated as (ST − CK)/CK × 100; multiple comparison was conducted by the Δ values; * and ** indicate significance at 0.05 and 0.01 among cultivars, respectively.

## 3.3. Photosynthetic Parameters

The photosynthetic parameters of all pomegranate cultivars irrigated with saline solution were negatively affected (Table 4), 27.19%~65.69% of Pn, 23.36%~66.72% of $g_s$, 1.71%~50.72% of Ci and 5.16%~45.95% of E, respectively. The higher reductions were found in "Al-sirin-nar", "Pecos", "Red Angel", "Surh-Anor", and "Sweet Peppermint".

**Table 4.** Photosynthetic parameter: leaf net photosynthesis (Pn), stomatal conductance (gs), intercellular CO2 concentration (Ci), and transpiration (E) of pomegranate leaves under salt stress.

| Cultivar | Pn ($CO_2$ μmol·m$^{-2}$·s$^{-1}$) | | | gs ($H_2O$ mmol·m$^{-2}$·s$^{-1}$) | | | Ci ($CO_2$ μmol·mol$^{-1}$) | | | E ($H_2O$ mmol·m$^{-2}$·s$^{-1}$) | | |
|---|---|---|---|---|---|---|---|---|---|---|---|---|
| | CK | ST | Δ (%) | CK | ST | Δ (%) | CK | ST | Δ (%) | CK | ST | Δ (%) |
| Al-sirin-nar | 10.32 | 3.54 | −65.69 ** | 74.40 | 38.30 | −48.52 * | 260.53 | 176.55 | −32.23 | 1.91 | 1.25 | −34.73 * |
| Dwarf Moy | 9.67 | 5.62 | −41.89 | 61.80 | 46.00 | −25.57 | 229.36 | 225.44 | −1.71 | 1.66 | 1.35 | −18.42 |
| Garnet Sash | 9.62 | 6.29 | −34.64 | 73.70 | 50.00 | −32.16 | 269.93 | 247.13 | −8.44 | 1.84 | 1.46 | −20.74 |
| Kandahar | 7.41 | 4.80 | −35.26 | 65.30 | 42.80 | −34.46 | 253.08 | 193.96 | −23.36 | 1.71 | 1.28 | −25.26 |
| Kara Bala Miursal | 8.22 | 4.06 | −50.59 | 83.20 | 39.40 | −52.64 * | 278.63 | 264.53 | −5.06 | 1.66 | 1.47 | −11.19 |
| Kara-Kalinskii | 8.47 | 3.78 | −55.33 * | 67.70 | 49.20 | −27.33 | 283.43 | 230.74 | −18.59 | 1.58 | 1.50 | −5.16 |
| Kazake | 9.40 | 5.62 | −40.25 | 61.70 | 43.10 | −30.15 | 252.53 | 144.66 | −42.72 * | 1.76 | 1.30 | −26.39 |
| Mollar | 8.36 | 4.67 | −44.15 | 107.20 | 69.90 | −34.79 | 279.64 | 250.15 | −10.55 | 2.18 | 1.62 | −25.59 |
| Pecos | 11.58 | 6.00 | −48.17 | 125.60 | 41.80 | −66.72 ** | 298.39 | 149.01 | −50.06 ** | 2.41 | 1.30 | −45.95 ** |
| Red Angel | 8.47 | 5.29 | −37.46 | 120.30 | 40.60 | −66.25 ** | 279.46 | 187.87 | −32.77 | 2.41 | 1.31 | −45.85 ** |
| Salavatski | 9.25 | 4.12 | −55.52 * | 69.00 | 37.90 | −45.07 * | 268.70 | 186.37 | −30.64 | 1.88 | 1.30 | −30.84 |
| Sirenevyi | 7.64 | 5.11 | −33.14 | 58.80 | 43.80 | −25.51 | 256.72 | 187.95 | −26.79 | 1.67 | 1.31 | −21.36 |
| Sogidavna | 7.80 | 5.68 | −27.19 | 61.00 | 45.20 | −25.90 | 274.51 | 135.29 | −50.72 ** | 1.66 | 1.32 | −20.46 |
| Surh-Anor | 8.60 | 3.98 | −53.70 * | 90.90 | 44.40 | −51.16 * | 290.33 | 214.90 | −25.98 | 2.05 | 1.31 | −36.168 |
| Sweet | 7.24 | 4.74 | −34.53 | 53.50 | 41.00 | −23.36 | 245.44 | 207.72 | −15.37 | 1.66 | 1.42 | −14.59 |
| Sweet Peppermint | 9.48 | 6.45 | −32.00 | 117.80 | 45.90 | −61.04 ** | 284.91 | 171.85 | −39.68 * | 2.44 | 1.35 | −44.75 ** |
| Vkusanyi | 7.13 | 4.16 | −41.68 | 88.40 | 61.70 | −30.20 | 307.22 | 259.38 | −15.57 | 1.99 | 1.51 | −24.15 |
| Wonderful | 11.09 | 6.88 | −37.96 | 84.00 | 44.50 | −47.02 * | 254.16 | 179.77 | −29.27 | 1.95 | 1.33 | −31.57 |

The values represented the means of 5 replications; CK: untreated group of plants, ST: salt-treated group of plants; Δ calculated as (ST − CK)/CK × 100; multiple comparison was conducted by the Δ values; * and ** indicate significance at 0.05 and 0.01 among cultivars, respectively.

### 3.4. Correlation and Cluster Analysis

The correlation analysis among 17 parameters was conducted using measured values of plants in treat group and control (Figure S2). The results showed that dry weight of roots, stems and leaves, total dry weight, root number, and new shoot number were positively correlated (indicated by blue color) with each other (correlation efficiency >0.60; $p < 0.05$). The leaf area, relative water content, Pn, $g_s$, Ci, E, root length and shoot length had significant positive correlation with each other ($p < 0.05$ and/or 0.01). Salt damage rate was negatively correlated to growth parameters (indicated by red color, Supplementary Figure S2).

According to the results of correlation analysis, 12 parameters were selected for cluster analysis, including total dry weight, root/shoot ratio, root length, root number, shoot length, new shoot number, leaf area, salt damage rate, relative water content, SPAD, Pn, and $g_s$. A dendrogram was developed with the selected 12 multivariate parameters of all pomegranate cultivars. Two clusters were identified (Figure 2). The cluster I (including "Salavatski", "Sirenevyi", "Kandahar", "Wonderful", "Kara Bala Miursal", "Kazake' 'Mollar", "Dwarf Moy", "Kara-Kalinskii", and "Vkusanyi") was more tolerant to NaCl stress than cluster II (including "Al-Sirin-nar", "Garnet sash", "Pecos", "Sogidavna", "Surh-Anor", "Sweet Peppermint", "Sweet", and "Red Angel"). The dendrogram was in line with salt damage rate, and 10 cultivars in cluster I showed lower salt damage rates (Figure 2).

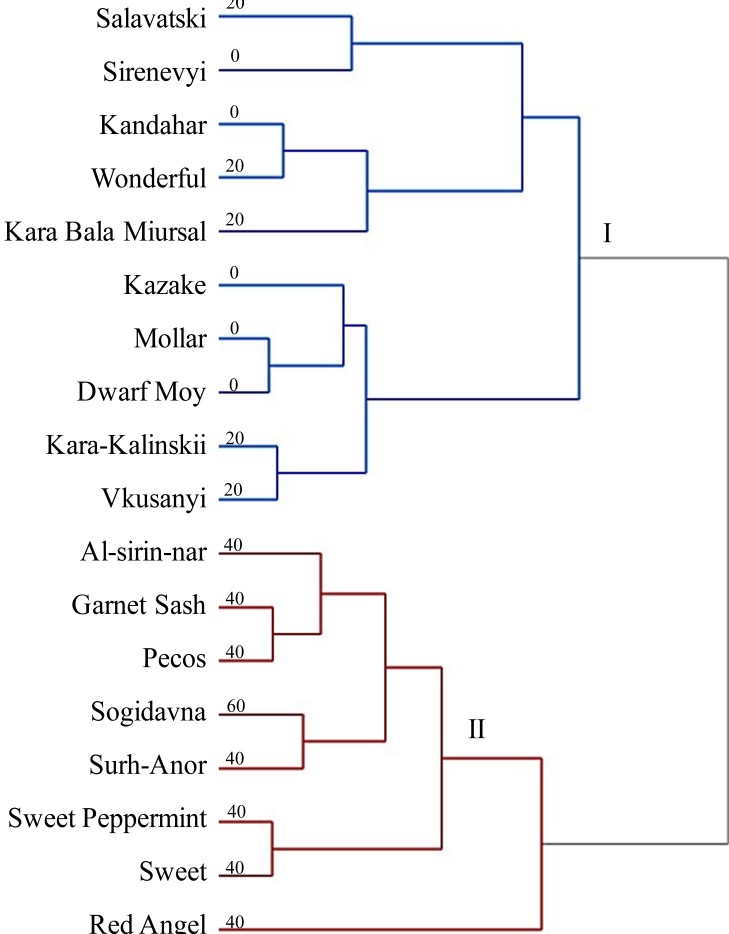

**Figure 2.** The dendrogram of cluster analysis of 18 pomegranate cultivars used the method of Ward linkage and squared Euclidian distance. It was based on difference values of plants between salt-treated group and control of 12 multivariate parameters, including total dry weight, root/shoot ratio, root length, root number, shoot length, new shoot number, leaf area, salt damage rate, relative water content, SPAD, Pn, and $g_s$. The numbers are foliar salt damage rate (%).

*3.5. Mineral Nutrients*

Four cultivars "Kandahar", "Dwarf Moy", "Mollar", and "Wonderful" in cluster I, and 4 cultivars "Red Angel", "Pecos", "Surh-Anor", and "Sweet" in cluster II were selected to determine the minerals content of roots, stems, and leaves. Under salt stress, Na concentration increased 1.0~6.10 times in roots, 1.37~9.70 times in stems, and 1.62~18.2 times in leaves, compared to control, respectively. Na concentration in roots were 1.72~9.91 times and 4.42~25.50 times more than that in stems and leaves of salt-treated plants (Table 5). Leaf P concentration in 8 cultivars all decreased, especially in "Pecos", "Dwarf Moy", "Mollar", and "Surh-Anor" (Table 5, $p < 0.05$ and/or 0.01). P concentration of roots and stems had no obvious change under salt stress (Table 5). P content of pomegranate organs and whole plant decreased under salt stress (Table S1). S concentration in pomegranate organs was ranked as root > leaf > stem, and it almost decreased under salt stress, when compared to control (Table 5). Root K concentration increased in pomegranate cultivars, except for "Red Angel" and "Surh-Anor", but stem K concentration decreased by 0.69%~15.64% compared to control. It was worth noting that leaf K concentration in "Kandahar", "Dwarf Moy", "Mollar", and "Wonderful" in cluster I increased, while it decreased in 4 cultivars in cluster II (Table 5). The reduction of K content of stems and leaves were caused by a decrease of dry weight (Table S1). Leaf Ca concentration in "Kandahar", "Dwarf Moy", "Mollar", and "Wonderful" in cluster I increased, while that of "Red Angel", "Pecos", "Surh-Anor", and "Sweet" in cluster II decreased. The total K content of whole plant decreased about 11.21%~34.28% (Table 5; Supplementary Table S1). Leaf Mg concentration of 8 cultivars mostly decreased under salt stress. No significant changes in stem Mg content were found. Total Mg content of whole plants decreased compared to control (Table 5; Supplementary Table S1).

The micronutrient concentration of pomegranate roots, stems, and leaves were determined, including Fe, Si, Al, Mn, Zn, and Cu (Table 6, Supplementary Table S2). Fe concentration significantly increased in roots of "Kandahar", "Dwarf Moy", "Mollar" and in stems of "Red Angel" and "Pecos". Leaf Fe concentration of 8 cultivars increased, especially in "Mollar", "Wonderful", and "Sweet" (Table 6, $p < 0.05$ or 0.01). Total Fe content of whole plant increased, except "Surh-Anor" (Supplementary Table S2). Si concentration and Si content of 8 cultivars mostly decreased under salt stress (Table 6, Supplementary Table S2). Most Mn concentration in leaves and stems increased, but it decreased in roots of "Red Angel", "Pecos", "Surh-Anor", and "Sweet" (Table 6). The content of Zn most decreased in pomegranate plants (Table 6, $p < 0.05$). Al mostly accumulated in roots, and the concentration increased in 4 cultivars in cluster I, but decreased in 4 cultivars in cluster II (Table 6). Cu concentration in roots, stems, and leaves of "Kandahar", "Dwarf Moy", "Mollar", and "Wonderful" all increased (except in root of "Wonderful"; Table 6). In total, the majority of cultivars had an increase in Fe, Mn, and Cu content and a decrease in Si, Al, and Zn content of whole plant under salt stress (Table S2).

**Table 5.** The accumulation of macroelements in pomegranate roots, stems, and leaves under salt stress.

| Concentration (mg/g) | | P | | | S | | | K | | | Ca | | | Na | | | Mg | | |
|---|---|---|---|---|---|---|---|---|---|---|---|---|---|---|---|---|---|---|---|
| Cultivar | Group | Root | Stem | Leaf | Root | Stem | Leaf | Root | Stem | Leaf | Root | Stem | Leaf | Root | Stem | Leaf | Root | Stem | Leaf |
| Dwarf Moy | CK | 3.32 | 3.60 | 5.47 | 5.13 | 1.60 | 2.67 | 8.73 | 15.00 | 20.40 | 7.80 | 9.73 | 12.67 | 4.00 | 1.87 | 0.20 | 2.33 | 1.60 | 4.07 |
|  | ST | 2.68 | 3.20 | 3.33 | 4.40 | 1.33 | 2.27 | 11.13 | 14.60 | 20.47 | 8.40 | 8.93 | 13.20 | 28.40 | 4.80 | 2.53 | 2.40 | 1.60 | 3.73 |
|  | Δ (%) | −16.25 | −11.11 | −39.02 * | −14.29 | −16.67 | −15.00 | 27.48 | −2.67 | 0.33 | 7.69 | −8.22 | 4.21 | 610 ** | 157 | 1167 * | 2.86 | 0.00 | −8.20 |
| Kandahar | CK | 2.13 | 2.73 | 4.40 | 4.73 | 1.33 | 2.67 | 10.07 | 13.60 | 16.07 | 10.53 | 9.87 | 13.87 | 6.73 | 0.93 | 0.33 | 3.20 | 1.33 | 3.60 |
|  | ST | 2.60 | 3.00 | 3.87 | 3.60 | 1.40 | 2.27 | 17.33 | 13.40 | 19.80 | 10.07 | 9.60 | 15.80 | 30.40 | 5.33 | 1.40 | 3.20 | 1.40 | 3.58 |
|  | Δ (%) | 21.87 | 9.76 | −12.12 | −23.94 | 5.00 | −15.00 | 72.19 ** | −1.47 | 23.24 * | −4.43 | −2.70 | 13.94 | 351 * | 471 * | 320 | 0.00 | 5.00 | −0.56 |
| Mollar | CK | 2.73 | 3.20 | 5.47 | 4.47 | 1.60 | 2.80 | 10.93 | 14.53 | 18.07 | 7.27 | 8.80 | 11.67 | 10.47 | 0.67 | 0.73 | 2.73 | 1.60 | 3.47 |
|  | ST | 2.40 | 2.60 | 3.40 | 4.20 | 1.20 | 2.20 | 14.00 | 14.43 | 22.60 | 8.07 | 8.20 | 12.00 | 21.60 | 7.13 | 2.33 | 2.33 | 1.60 | 3.40 |
|  | Δ (%) | −12.20 | −18.75 | −37.80 * | −5.97 | −25.00 | −21.43 | 28.05 | −0.69 | 25.09 * | 11.01 | −6.82 | 2.86 | 106 | 970 ** | 218 | −14.63 | 0.00 | −1.92 |
| Wonderful | CK | 2.67 | 3.13 | 6.40 | 5.00 | 1.80 | 3.73 | 9.80 | 14.53 | 20.02 | 9.73 | 9.53 | 11.40 | 3.93 | 0.80 | 0.27 | 2.27 | 1.40 | 3.40 |
|  | ST | 3.00 | 4.00 | 4.53 | 3.40 | 1.60 | 2.40 | 12.33 | 14.27 | 20.73 | 8.27 | 7.67 | 11.87 | 24.93 | 4.33 | 1.73 | 2.60 | 1.40 | 2.93 |
|  | Δ (%) | 12.50 | 27.66 | −29.17 | −32.00 * | −11.11 | −35.71 * | 25.85 | −1.83 | 3.56 | −15.07 | −19.58 | 4.09 | 534 ** | 442 * | 550 | 14.71 | 0.00 | −13.73 |
| Pecos | CK | 4.00 | 4.40 | 6.53 | 4.73 | 2.00 | 3.60 | 12.27 | 16.73 | 20.00 | 6.67 | 10.07 | 10.80 | 10.00 | 0.80 | 0.17 | 2.40 | 1.60 | 3.27 |
|  | ST | 3.07 | 3.13 | 2.93 | 4.07 | 1.40 | 2.33 | 13.27 | 14.33 | 17.33 | 6.60 | 9.67 | 10.53 | 23.40 | 5.20 | 3.20 | 2.33 | 1.80 | 2.93 |
|  | Δ (%) | −23.33 | −28.79 | −55.10 * | −14.08 | −30.00 | −35.19 * | 8.15 | −14.34 | −13.33 | −1.00 | −3.97 | −2.47 | 134 | 550* | 1820 ** | −2.78 | 12.50 | −10.20 |
| Red Angel | CK | 2.93 | 3.00 | 4.33 | 6.40 | 1.47 | 2.73 | 13.87 | 16.33 | 21.60 | 10.20 | 8.27 | 12.47 | 5.53 | 1.07 | 0.53 | 4.13 | 1.13 | 3.40 |
|  | ST | 2.80 | 3.40 | 4.20 | 4.07 | 1.40 | 2.27 | 10.80 | 13.80 | 17.60 | 7.67 | 7.47 | 10.27 | 20.07 | 2.53 | 1.40 | 2.20 | 1.20 | 2.80 |
|  | Δ (%) | −4.55 | 13.33 | −3.08 | −36.46 * | −4.55 | −17.07 | −22.12 | −15.51 | −18.52 | −24.84 * | −9.68 | −17.65 | 262 | 137 | 162 | −46.77 ** | 5.88 | −17.65 |
| Surh-Anor | CK | 2.87 | 3.53 | 5.07 | 4.67 | 1.67 | 2.80 | 11.80 | 16.60 | 17.67 | 8.13 | 7.73 | 11.87 | 12.00 | 0.67 | 0.33 | 2.87 | 1.40 | 3.67 |
|  | ST | 2.27 | 2.60 | 3.07 | 3.73 | 1.00 | 2.40 | 11.67 | 15.00 | 17.13 | 7.53 | 6.13 | 10.53 | 24.07 | 4.60 | 2.00 | 2.73 | 1.40 | 3.20 |
|  | Δ (%) | −20.93 | −26.42 | −39.47 * | −20.00 | −40.00 * | −14.29 | −1.13 | −9.64 | −3.02 | −7.38 | −20.69 | −11.24 | 100 | 590 * | 500 | −4.65 | 0.00 | −12.73 |
| Sweet | CK | 2.73 | 3.60 | 6.33 | 6.40 | 1.33 | 2.80 | 12.73 | 16.20 | 21.93 | 8.13 | 8.80 | 12.47 | 5.00 | 1.47 | 0.07 | 3.87 | 1.40 | 3.53 |
|  | ST | 2.93 | 3.07 | 4.87 | 4.00 | 1.40 | 2.87 | 12.93 | 13.67 | 19.27 | 8.67 | 9.40 | 11.00 | 25.40 | 3.67 | 0.87 | 2.53 | 1.40 | 3.47 |
|  | Δ (%) | 7.32 | 14.72 | −23.16 | −37.50 * | 5.00 | 2.38 | 1.57 | −15.64 | −12.16 | 6.56 | 6.82 | −11.76 | 408 * | 150 | 1200 * | −34.48 ** | 0.00 | −1.89 |

The values represented the means of 5 replications; CK: untreated group of plants, ST: salt-treated group of plants; Δ calculated as (ST − CK)/CK × 100; multiple comparison was conducted by the Δ values; * and ** indicate significance at 0.05 and 0.01 among cultivars.

**Table 6.** The accumulation of microelements in pomegranate roots, stems and leaves under salt stress.

| Concentration (µg/g) | | Fe | | | Si | | | Mn | | | Zn | | | Al | | | Cu | | |
|---|---|---|---|---|---|---|---|---|---|---|---|---|---|---|---|---|---|---|---|
| Cultivar | Group | Root | Stem | Leaf | Root | Stem | Leaf | Root | Stem | Leaf | Root | Stem | Leaf | Root | Stem | Leaf | Root | Stem | Leaf |
| **Dwarf Moy** | CK | 0.48 | 0.20 | 0.24 | 0.20 | 0.01 | 0.27 | 0.07 | 0.08 | 0.14 | 0.12 | / | / | 0.36 | / | / | 0.02 | 0.03 | 0.02 |
| | ST | 0.60 | 0.20 | 0.46 | 0.20 | 0.03 | 0.26 | 0.13 | 0.09 | 0.22 | 0.10 | / | / | 0.43 | / | / | 0.07 | 0.15 | 0.14 |
| | Δ (%) | 23.83 | −3.92 | 90.64 | −1.65 | 137 * | −1.50 | 102 * | 12.60 | 56.41 | −10.85 | / | / | 17.58 | / | / | 260 * | 660 ** | 620 ** |
| **Kandahar** | CK | 0.32 | 0.17 | 0.03 | 0.20 | 0.06 | 0.14 | 0.06 | 0.02 | 0.05 | 0.06 | 0.04 | 0.02 | 0.37 | 0.08 | / | 0.01 | 0.02 | 0.02 |
| | ST | 0.46 | 0.18 | 0.15 | 0.22 | 0.06 | 0.17 | 0.17 | 0.05 | 0.10 | 0.07 | 0.03 | 0.01 | 0.39 | 0.03 | / | 0.11 | 0.15 | 0.13 |
| | Δ (%) | 43.41 * | 5.88 | 460 ** | 12.96 | 3.45 | 20.66 | 213 ** | 134 * | 104* | 18.85 | −20.77 | −54.13 | 5.95 | −56.41 | / | 685 ** | 630 ** | 540 ** |
| **Mollar** | CK | 0.37 | 0.09 | 0.07 | 0.22 | 0.05 | 0.07 | 0.08 | 0.04 | 0.07 | 0.04 | 0.07 | 0.04 | 0.35 | 0.02 | / | 0.02 | 0.02 | 0.03 |
| | ST | 0.53 | 0.09 | 0.16 | 0.26 | 0.06 | 0.09 | 0.21 | 0.08 | 0.10 | 0.05 | 0.07 | 0.01 | 0.49 | 0.02 | / | 0.08 | 0.15 | 0.23 |
| | Δ (%) | 41.29 * | −4.48 | 147 | 17.85 | 16.44 | 24.54 | 168 ** | 92.29 * | 52.94 | 35.11 | 4.91 | −68.65 | 41.95 | 10.00 | / | 320 * | 640 ** | 1052 ** |
| **Wonderful** | CK | 0.56 | 0.12 | 0.11 | 0.18 | / | 0.26 | 0.08 | 0.05 | 0.11 | 0.10 | 0.02 | 0.01 | 0.40 | / | / | / | 0.02 | 0.02 |
| | ST | 0.54 | 0.15 | 0.30 | 0.20 | / | 0.22 | 0.17 | 0.08 | 0.11 | 0.12 | 0.01 | 0.03 | 0.42 | / | / | / | 0.17 | 0.13 |
| | Δ (%) | −4.05 | 30.51 | 177 * | 16.35 | / | −17.93 | 114 * | 68.52 | 5.52 | 18.84 | −100 | 299** | 3.98 | / | / | / | 740 ** | 550 ** |
| **Pecos** | CK | 0.52 | 0.10 | 0.06 | 0.19 | 0.01 | 0.14 | 0.23 | 0.03 | 0.08 | 0.16 | 0.03 | 0.04 | 0.47 | / | / | / | / | / |
| | ST | 0.54 | 0.22 | 0.21 | 0.19 | 0.03 | 0.21 | 0.16 | 0.07 | 0.10 | 0.05 | 0.02 | 0.02 | 0.31 | / | / | / | / | / |
| | Δ (%) | 4.65 | 120 ** | 253 * | 0.87 | 193.40 * | 57.08 | −30.65 | 90.42 * | 26.87 | −68.22 | −86.95 | −100 | −34.04 | / | / | / | / | / |
| **Red Angel** | CK | 0.41 | 0.16 | 0.23 | 0.20 | 0.04 | 0.11 | 0.11 | 0.11 | 0.18 | 0.06 | 0.03 | / | 0.31 | 0.02 | / | 0.02 | / | / |
| | ST | 0.46 | 0.26 | 0.37 | 0.21 | 0.09 | 0.10 | 0.14 | 0.10 | 0.18 | 0.05 | 0.03 | / | 0.30 | 0.05 | / | 0.02 | / | / |
| | Δ (%) | 13.96 | 67.95 * | 63.92 | 4.03 | 120 * | −7.32 | 33.53 | −10.53 | −3.28 | −13.26 | −11.33 | / | −3.25 | 150 ** | / | 11.11 | / | / |
| **Surh-Anor** | CK | 0.45 | 0.11 | 0.10 | 0.21 | 0.01 | 0.26 | 0.13 | 0.04 | 0.07 | 0.06 | / | 0.01 | 0.39 | / | / | / | / | 0.02 |
| | ST | 0.40 | 0.16 | 0.14 | 0.22 | 0.04 | 0.22 | 0.10 | 0.07 | 0.10 | 0.04 | / | 0.01 | 0.35 | / | / | / | / | 0.05 |
| | Δ (%) | −12.65 | 40.70 | 33.78 | 7.12 | 313 ** | −15.94 | −16.94 | 87.29 * | 32.42 | −38.05 | / | −100 | −11.17 | / | / | / | / | 160 |
| **Sweet** | CK | 0.64 | 0.17 | 0.13 | 0.17 | 0.01 | 0.17 | 0.17 | 0.05 | 0.10 | 0.16 | 0.03 | 0.02 | 0.47 | / | / | / | / | / |
| | ST | 0.58 | 0.23 | 0.34 | 0.21 | 0.01 | 0.23 | 0.16 | 0.07 | 0.14 | 0.09 | 0.02 | 0.01 | 0.40 | / | / | / | / | / |
| | Δ (%) | −8.17 | 33.99 | 151 | 25.70 | 22.17 | 37.65 | −2.37 | 38.96 | 37.25 | −41.64 | −100 | −100 | −14.10 | / | / | / | / | / |

The values represented the means of 5 replications; CK: untreated group of plants, ST: salt-treated group of plants; Δ calculated as (ST − CK)/CK × 100; multiple comparison was conducted by the Δ values;/was less than 0.01 µg/g and unavailable; * and ** indicate significance at 0.05 and 0.01 among cultivars.

## 4. Discussion

### 4.1. Growth and Physiological Parameters of Pomegranate Cultivars

The growth feature is the most important and intuitive indication to assess plant salt tolerance. Our findings show that high salinity (200 Mm NaCl) impacts pomegranate growth negatively, with a large variation among cultivars. These results were in line with other previous works that increasing salinity level will inhibit pomegranate growth in terms of shoot length, leaf area, dry weight, flowering, or yield [14,23]. Dry weight of 18 pomegranate cultivars decreased, length and number of shoot and root, as well as leaf area were negatively affected after 35 days of 200 mM NaCl treating. Inhibition of shoot and root development is the primary response to the stress [24]. Restriction of root growth by salinity reduced the uptake of water and essential minerals, diminished supply of water and nutrients to shoot, which might contribute to growth reduction [24]. An increased root/shoot ratio was observed in 11 pomegranate cultivars under salt stress, which was mainly due to the higher sensitivity of shoot than root to salt stress [24]. The lower RWC of NaCl-treated pomegranate leaves reflected the impairment in the capacity of water retention [25], which caused stomatal closure, then reduced the rate of photosynthesis and transpiration. A potentially negative effect of reduction in Pn and leaf area was the decreased total $CO_2$ assimilation and thus inhibited plant growth and development under saline conditions [24,26]. Consequently, the limited vegetative growth resulted in a smaller pomegranate plant with shorter shoots. However, there were a few occurrences of leaf burn, necrosis, or discoloration in some pomegranate cultivars (Figure 1). Sun et al. [14] also found no visual foliar salt damage on pomegranate plants during their entire experimental period. We found that pomegranate is relatively tolerant to salt stress [10,12,13], contrary to Bhantana and Lazarovitch [2]. Based on cluster analysis, ten cultivars including "Salavatski", "Sirenevyi", "Kandahar", "Wonderful", "Kara Bala Miursal", "Kazake" "Mollar", "Dwarf Moy", "Kara-Kalinskii" and "Vkusanyi" were considered as salt tolerant cultivars.

### 4.2. Effects of NaCl Stress on Ion Content of Pomegranate Tissues

In saline environment, plants firstly experience osmotic stress, and then suffer from ion toxicity and nutrients deficiency due to excessive salt ions in leaves after weeks or months [27]. Plant potential responses to salinity are associated with the ability of nutrients uptake and/or transport into plant tissues [28,29]. Previous studies have reported that a high NaCl stress mainly resulted in an over accumulation of Na and Cl [30], and even other mineral ions are imbalance at both the cellular and whole plant levels [9,31]. Our study showed that Na content of 8 pomegranate cultivars increased sharply under salt stress, and the Na concentrations of roots were much more than that of stems and leaves (1.72~9.91 times and 4.42~25.50 times; Table 5). After 35 days of 200 mM NaCl stressing, plants of the 13 cultivars had very mild leaf burn and discoloration, which was probably due to an ability to accumulate more Na in roots and transport less Na to leaves [32]. Hence, we speculate that it is a strategy for pomegranate to alleviate the detrimental effects of salt stress. Plant salt-resistance has been linked to the retention of salt ions in roots [33] and/or preventing the accumulation of Na in shoots and leaves in other researches [34].

Macronutrients are critical structural components of plants, and their deficiency may significantly affect plant growth and development [35]. The decrease of P and S concentration in most cultivars suggested the uptake of P and S were inhibited by salt stress [36]. More K and Ca accumulated in leaves than in roots for all cultivars (Table 5; Supplementary Table S1). When excess Na accumulated in root cells, it would compete for K and Ca binding sites, both K and Ca might be replaced by Na and then transported into shoots during transpiration [5,37]. We also calculated the ratios of K/Na and Ca/Na in pomegranate organs (Supplementary Table S3). K/Na and Ca/Na ratios of stems and leaves decreased significantly, but the values were higher than 1 under 200 mM NaCl stress (Supplementary Table S3). The K/Na and Ca/Na are confirmed as key determinants of plant salt resistance [5,36]. A K/Na or Ca/Na ratio of approximately 1 is the minimum value for normal metabolism [38]. On the other hand,

an increased concentration of root K, leaf K, and leaf Ca in relatively tolerant cultivars "Kandahar", "Dwarf Moy", "Mollar", and "Wonderful" was observed (Table 5). Also, a large decrease of leaf K and leaf Ca content of relatively sensitive cultivars "Red Angel", "Pecos", "Surh-Anor", and "Sweet" was found (Supplementary Table S1). Our results implied that some pomegranate cultivars had a higher ability of K and Ca transport into shoots and leaves, and then maintained a suitable K/Na or Ca/Na ratio for normal metabolism [31,39,40]. These results were similar to those on oak (*Quercus virginiana*) [37] and chickpea (*Cicer arietinum*) [41] in saline conditions, in which it was reported that the higher capacity for K and Ca transport to the aerial part would contribute to mitigating the ion toxicity in leaf cells. Mg serves as a chlorophyll component and activator involved in photosynthesis, and the decrease of leaf Mg concentration might be one reason for photosynthesis impairment (Table 5) [42]. Nearly no change was observed in the stem Mg content of 8 pomegranate cultivars under salt stress, which was similar to results in guava (*Psidium guajava*) [43].

Micronutrients directly or indirectly affect the activity of catalytic enzymes and metabolites involving in plant responses to abiotic stresses [29]. Despite their low concentrations within the plant tissues and organs, micronutrients are of equal importance to macronutrients for plants [28]. Many studies had showed that pomegranate cultivars had different abilities for micronutrients uptakes from soil [44,45]. Hasanpour et al. [15] reported that Zn, Cu, and Mn concentration of in "Rabab" pomegranate roots increased, but root Fe decreased with increasing salinity. Khayyat et al. [46] found leaf Fe decreased in "Malas–e–Saveh" and "Shishe–Kab" Iranian pomegranates, while leaf Zn concentration increased by salinity up to 9 dS·m$^{-1}$ and then decreased. Tester and Davenport [47] stated that salinity decreased Fe and Zn uptake from the soil solution via reducing root growth and development. In general, most Fe, Mn and Cu concentration and total content of the whole plant increased in this experiment, which was similar to the results of soybean (*Glycine max*) [48] and mango (*Mangifera indica*) [49]. While Al, Si, and Zn content of plant mostly decreased under salt stress, with a large variation among cultivars (Table 6; Supplementary Table S2). These results indicated that salt stress caused Al, Si, and Zn deficiency in pomegranate plant.

The correlation between growth parameters and total mineral content was analyzed (Figure S3). The results showed that Na content was negatively correlated to Pn, RWC, K/Na, and Ca/Na, which indicated physiological activity and ion balance were closely negatively related to superoptimal Na amount in cell. Salt damage rate was negatively correlated with P, S, K, Ca, Mg, Si and Al. These findings suggested that salt stress negatively impacted mineral absorption [29]. Plants with a higher capacity for mineral absorption possess a more efficient metabolism cycle and better adaptability to environmental conditions [27]. As showed in Tables 5 and 6, four relatively tolerant cultivars have higher concentrations of K, Ca, Fe, Mn, and Cu in leaves and roots (except Ca) than four relatively sensitive cultivars. We speculate that some pomegranate cultivars have a higher tolerance to salinity than the others by selectively absorbing more mineral ions from soils [11,18].

## 5. Conclusions

Salt treatment impacted pomegranate growth negatively, with a large variation among the 18 cultivars investigated in this study. However, only very minor foliar damage was observed on 13 cultivars during the entire experiment. Pomegranate is tolerant to salt stress to some extent, which might be due to the higher ability of storing Na in roots and transporting K and Ca to leaves. This might be a strategy for pomegranate to cope with salt stress, but it needs further study on various salinity level and stress duration. Ten relative-tolerant cultivars ("Salavatski", "Sirenevyi", "Kandahar", "Wonderful", "Kara Bala Miursal", "Kazake" "Mollar", "Dwarf Moy", "Kara-Kalinskii", and "Vkusanyi") are selected to offer the reference for pomegranate cultivation on saline lands. Future research to quantify the effect of salt stress on yield and quality is needed.

**Supplementary Materials:** The following are available online at http://www.mdpi.com/2073-4395/10/1/27/s1, Figure S1: The images of software - Easy leaf area on phone, Figure S2: The correlation efficiency among growth parameters, Figure S3: The correlation efficiency between growth parameters and mineral nutrients,

Table S1: The accumulation of macroelements in pomegranate roots, stems and leaves under salt stress, Table S2: The accumulation of microelements in pomegranate roots, stems and leaves under salt stress, Table S3: The ratio of K/Na and Ca/Na in pomegranate organs.

**Author Contributions:** Conceptualization, C.L., Z.Y. and M.G.; methodology C.L.; formal analysis, C.L. and X.Z.; investigation, C.L. and J.Y.; writing—original draft preparation, C.L.; writing—review and editing, C.L., X.Z., Z.H. and M.G.; funding acquisition, Z.Y. Supervision, M.G. All authors have read and agreed to the published version of the manuscript.

**Funding:** This work was supported by the Initiative Project for Talents of Nanjing Forestry University (GXL2014070, GXL2018032), the Natural Science Foundation of Jiangsu Province (BK20180768), the Doctorate Fellowship Foundation of Nanjing Forestry University, and the Priority Academic Program Development of Jiangsu High Education Institutions (PAPD).

**Conflicts of Interest:** The authors declare no conflict of interest.

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
