# Peer review of "Effects of Salt Stress on Growth, Photosynthesis, and Mineral Nutrients of 18 Pomegranate (Punica granatum) Cultivars"

_agronomy, doi:10.3390/agronomy10010027_

Round 1
Reviewer 1 Report
Reviewer’s comments to the authors
I reviewed a research article manuscript with a topic on the effect of salinity on growth, photosynthetic parameters and nutrient content of 18 pomegranate varieties. Generally this research is well organized, the results are clear and sound and the interpretation of the data is also good. One point that raised my attention is that as far as this is an applied research experiment the concentration of the salt treatment (200mM) is not that ‘’realistic’’ in most of the exiting soils abroad (maybe trying 50 or 100 mM would be more) but nevertheless I understand that such a high concentration would give anyway faster stress symptoms to the plants. However this work is interesting but I strongly believe the authors should consider English editing from a native speaker as there are several grammar or syntax mistakes in the whole text.
Above please find some more detailed comments:
P1L25: rephrase sentence (may delete ‘’with’’)
P1L26-28: this sentence refers to a speculation or what? The abstract should contain only data obtained from the work. May rephrase the sentence in order to look more like results of the work.
P2L74: I can’t understand what authors call ‘’treatment plants’’ are the plants subjected to the salinity treatment maybe?. Please rephrase the syntax here is wrong.
P2L88: provide the abbreviation of SPAD (abbreviations should be given in full the first time shown in the text)
P3L118: the authors described an experiment with 5 blocks little earlier. Why in the end they conducted statistical analysis with 5 replicates and not with 5?
P4L153: ‘’ showed gist to count’’. Please try to be more descriptive what is observed in this figure while please reconsider the use of English language. Furthermore, figures subtitles should be such explanatory so that they can stand even alone.
P7L162: replace ‘’little’’ with ‘’some’’ or another word…
P14L218: may consider using the same time verb in whole text when providing your data. Instead of has use had?
P12L244: a verb is missing to the second part of the sentence (after while…)
P12L248: add ‘the’ majority
P14L244: replace accumulations with accumulation (in all tables also in supplementary)
P13L273: rephrase ‘’slowed’’ is false used
P14L311: ‘’more decrease’’ the English use is wrong rephrase
Author Response
Dear reviewer,
The authors would like to thank you for your time and effort in improving this manuscript. The following is a point-by-point response to each comment.
Please let me know if you have any questions.
Sincerely,
Cuiyu Liu

Reviewer 2 Report
While the work is not particularly original from a hypothesis driven viewpoint, it is a well- structured descriptive -response study that provides substantial information on response of many pomegranate cultivars to a controlled salinity induced study. The experimental design is sound and the results provide good information on a number of cultivars that shows a range of response to salinity stress. The abundant data provided are potentially useful for extension recommendations for growers, as well as for supporting physiological impacts on growth and development. The findings may be useful for additional studies, and advancement of current knowledge on salinity stress in pomegranate.
Interpretation of the data is valid and justified. The manuscript requires considerable revision. Plural and singular grammatical structure needs more attention during revision. I have listed many examples for revision and undoubtedly, much more editing is required to improve the presentation.
The work appears to be scientifically sound. The paper is a nice example of how a mostly descriptive, observational (non-hypothesis driven) project can be presented with relevance. The choice of a large number of known cultivars improves the scope of the work.
Readers with extension, teaching, and additional research should be interested. Thus the work should be published if, this journal accepts well-done descriptive research results. The English presentation must be improved.
The data on responses to controlled salinity stress for a large array of tolerant and less-tolerant cultivars is convincing, and the large amount of data in tables on changes in macro and micronutrients provide a sound contribution to the literature for practical and future research applications.
The section on photosynthetic response is less useful. It is a cursory treatment that is not particularly well-related to the main body of the work.
Author Response
Dear Reviewer,
The authors would like to thank you for your time and effort in improving this manuscript. The following is a point-by-point response to each comment.
Please let me know if you have any questions.
Sincerely,
Cuiyu Liu
Point 1: I have listed many examples for revision and undoubtedly, much more editing is required to improve the presentation. The English presentation must be improved.
Response 1: We have revised based on your suggestion. And we had an English editing on our paper.
Point 2: The section on photosynthetic response is less useful. It is a cursory treatment that is not particularly well-related to the main body of the work.
Response 2: The photosynthesis change is an important physiological response to salt stress, which is well-related to plant growth. We also used photosynthetic parameters to analysis the correlation with other variables.
Reviewer 3 Report
Dear Editor and Authors, this is a well conducted study that provides many information about salt tolerance of pomegranate. There are only minor corrections needed. Please check pdf annotations in uploaded file. Also:
1) Some strange expressions in lines 259-260 of the Discussion should be revised.
2) Please provide the total electrical conductivity of the nutrient solutions supplied in the control and salt-treated plants. This is needed for easy reference and comparison with results of other studies.

Author Response
Dear Reviewer,
The authors would like to thank you for your time and effort in improving this manuscript. The following is a point-by-point response to each comment.
Please let me know if you have any questions.
Sincerely,
Cuiyu Liu
